# Effect of Dietetic Obesity on Testicular Transcriptome in Cynomolgus Monkeys

**DOI:** 10.3390/genes14030557

**Published:** 2023-02-23

**Authors:** Yanru Zhang, Jia Qi, Juan Zhao, Miaojing Li, Yulin Zhang, Huizhong Hu, Liangliang Wei, Kai Zhou, Hongyu Qin, Pengxiang Qu, Wenbin Cao, Enqi Liu

**Affiliations:** 1Laboratory Animal Center, Xi’an Jiaotong University Health Science Centre, Xi’an 710061, China; 2Department of Hematology, The First Affiliated Hospital of Xi’an Jiaotong University, Xi’an 710061, China; 3Precision Medicine Center, The First Affiliated Hospital of Xi’an Jiaotong University, Xi’an 710061, China; 4Key Laboratory of Environment and Genes Related to Diseases, Ministry of Education of China, Xi’an 710049, China

**Keywords:** testis, obesity, transcriptome, cynomolgus monkey

## Abstract

Obesity is a metabolic disorder resulting from behavioral, environmental and heritable causes, and can have a negative impact on male reproduction. There have been few experiments in mice, rats, and rabbits on the effects of obesity on reproduction, which has inhibited the development of better treatments for male subfertility caused by obesity. Nonhuman primates are most similar to human beings in anatomy, physiology, metabolism, and biochemistry and are appropriate subjects for obesity studies. In this investigation, we conducted a transcriptome analysis of the testes of cynomolgus monkeys on high-fat, high-fructose, and cholesterol-rich diets to determine the effect of obesity on gene expression in testes. The results showed that the testes of obese monkeys had abnormal morphology, and their testes transcriptome was significantly different from that of non-obese animals. We identified 507 differentially abundant genes (adjusted *p* value < 0.01, log2 [FC] > 2) including 163 up-regulated and 344 down-regulated genes. Among the differentially abundant genes were ten regulatory genes, including *IRF1*, *IRF6*, *HERC5*, *HERC6*, *IFIH1*, *IFIT2*, *IFIT5*, *IFI35*, *RSAD2*, and *UBQLNL*. Gene ontology (GO) and KEGG pathway analysis was conducted, and we found that processes and pathways associated with the blood testes barrier (BTB), immunity, inflammation, and DNA methylation in gametes were preferentially enriched. We also found abnormal expression of genes related to infertility (*TDRD5*, *CLCN2*, *MORC1*, *RFX8*, *SOHLH1*, *IL2RB*, *MCIDAS*, *ZPBP*, *NFIA*, *PTPN11*, *TSC22D3*, *MAPK6*, *PLCB1*, *DCUN1D1*, *LPIN1*, and *GATM*) and down-regulation of testosterone in monkeys with dietetic obesity. This work not only provides an important reference for research and treatment on male infertility caused by obesity, but also valuable insights into the effects of diet on gene expression in testes.

## 1. Introduction

Obesity is marked by an excessive accumulation of body fat that presents a risk to health and has reached a pandemic level in the world [1]. According to a recent report from the World Health Organization, more than 1.9 billion adults aged 18 and above were overweight (body mass index (BMI) ≥ 25 kg/m^2^), of which more than 650 million were obese (BMI ≥ 30 kg/m^2^), and among them, 39% of men are overweight and 11% are obese [1,2]. Currently, the increase in the number of obese individuals coincides with the increase in male infertility, which threatens the reproduction ability of thousands of couples and the health of their offspring [3,4].

Obese men are more likely to have decreased sperm concentration, and to have sperm with abnormal morphology, damaged chromatin, and reduced motility [5,6]. BMI is closely and negatively correlated with sperm quality parameters, and obese men are three times as likely to have abnormal sperm parameters as normal men [7]. Sperm from obese men show higher levels of reactive oxygen species (ROS) and DNA damage [8]. Paternal obesity can result in embryos with abnormal cleavage, abnormal gene expression, and lower developmental competence [9]. The offspring of obese fathers also have an increased risk of metabolic disease, and a high-fat diet can change the DNA methylation, histone modification, transcriptome, proteome, etc., of gametes, and some obesity factors can be inherited through the sperm [10,11,12].

At present, a consensus has been reached on the adverse effects of obesity from a high-fat or high-sugar diet on male reproduction. Non-human primates, as the animals closest to humans, can be used as important disease models to study human metabolic diseases [13]. Up to now, there have been few reports on the testicular transcriptome in non-human primates. In this study, we conducted a transcriptome analysis of the testes of cynomolgus monkeys on a diet high in fat, fructose, and cholesterol, which is similar to the modern Western pattern diet likely to lead to obesity. Our goal was to reveal differentially abundant genes in testis related to obesity to provide a reference for the prevention and cure of male sterility.

## 2. Materials and Methods

### 2.1. Animals

All experimental protocols involving nonhuman primates were approved by the Laboratory Animal Care Committee of Xi’an Jiaotong University. Cynomolgus monkeys (*Macaca fascicularis*) were provided by Spring Biological Technology Development Co., Ltd. (Fangchenggang, China). Ten male monkeys (age ≥ 9 years) were used in this study, five in the control group, which were fed with normal chow diet, and five in the obesity group, which were fed a diet high in fat, fructose, and cholesterol. The normal chow diet was made according to the Nutrient Requirements of Nonhuman Primates (https://www.nature.com/articles/laban1103-26, accessed on 4 June 2019). The high-fat diet was purchased from Beijing Keao Xieli Feed Co., Ltd. (www.keaoxieli.com, accessed on 4 June 2019) with 24% fat, 15% fructose, and 1% cholesterol. Both groups of monkeys were fed for 15 months. Afterward, monkeys with BMI > 42 were assessed as overweight. The temperature of the animal room was 22–25 °C, the humidity was 48–65%, and the light time was 12 h a day. All monkeys were housed in individual cages with access to water and food ad libitum. In the end, the monkeys were anesthetized and then treated with bloodletting. The testes were quickly collected, and both sides were placed in liquid nitrogen or 4% paraformaldehyde.

### 2.2. Hematoxylin and Eosin (H & E) Staining of Testes

Testes were fixed in 4% paraformaldehyde for 24 h, and then placed in a dehydrator for dehydration, cleared, immersed in wax, and embedded in paraffin blocks. The tissues were cut into 5 µm-thick sections and dried. The sections were dewaxed in xylene, then in 100%, 95%, and 90% alcohol, successively, and rinsed in Deionized (DI) water for 3 min. They were stained with H & E then immersed in 90%, 95%, and 100% alcohol for dehydration, and lastly in two changes of xylene, and were mounted using neutral resin.

### 2.3. Assay for Testicular Testosterone

About 200 mg of testicular tissue, the middle portion of the left testis, was weighed and rinsed with pre-chilled phosphate-buffered saline (PBS) to remove residual blood. PBS was added at a weight-to-volume ratio of 1:9; the tissue samples were treated on ice for 5 min with TissueMaster™ Handheld Homogenizer (Beyotime Biotechnology, Shanghai, China), and the homogenates were centrifuged at 5000× *g* for 10 min to collect the supernatant. The monkey testosterone enzyme-linked immunosorbent assay kit (Enzyme Linked Biotechnology, Shanghai, China) was used to assay testosterone. The coated plate was removed after equilibrating at room temperature for 1 h. Standard wells were added with 50 μL different concentrations of standards, and blank and sample wells were added with 50 μL sample diluents and samples, respectively. Then, 100 μL horseradish peroxidase (HRP)-labeled antibody was added to each well, and the plate was covered and incubated in the 37 °C incubator for 1 h. After that, the plate cover was removed, and the contents of the microtiter wells were thoroughly aspirated after washing each well five times with 350 μL washing buffer. Subsequently, the plate was incubated at 37 °C for 15 min (without direct exposure to intense light) after the addition of 50 μL substrates A and B to each well. Finally, the absorbance (OD) of each well was measured at 450 nm after adding 50 μL stop solution to each well.

### 2.4. RNA-Sequencing

Total RNA was extracted from tissues using TRIzol (Invitrogen, Carlsbad, CA, USA) following the company’s instructions. After weighing out 60 mg of tissue, it was flash-frozen in liquid nitrogen and ground to powder. The powder was transferred to a 2 mL centrifuge tube, dissolved in 1.5 mL TRIzol, and centrifuged at 12,000× *g* for 5 min at 4 °C. The supernatants were transferred to another 2 mL centrifuge tube and mixed with 0.3 mL of chloroform/isoamyl alcohol. After shaking for a few seconds, the extracts were centrifuged at 12,000× *g* for 10 min at 4 °C, and the supernatants were transferred to clean 1.5 mL centrifuge tubes. Equal volumes of isopropanol were then added to the transferred liquid, mixed, and centrifuged at 12,000× *g* for 20 min at 4 °C. The supernatants were removed and discarded, and the precipitated RNA was rinsed with 1 mL of 75% ethanol. Samples were placed in a biosafety cabinet for air-drying, and then dissolved in 25 µL to 100 µL of diethyl pyrocarbonate (DEPC)-treated water. RNA quality was checked by A260/280 using a NanoDrop spectrometer and concentration was determined with an Agilent 2100 bioanalyzer (Thermo Fisher Scientific Waltham, MA, USA).

Ribosomal RNA (rRNA) was isolated using target-specific oligos. After solid-phase reversible immobilization (SPRI) selection, the RNA was separated into multiple small fragments under the treatment of divalent cations. The separated RNA pieces were copied into first-strand cDNA with reverse transcriptase and random primers, followed by second-strand cDNA synthesis using DNA polymerase and Ribonuclease H (RNase H). The RNA template was digested by this method; a fresh substitute strand was synthesized, and double-stranded cDNA was generated by replacing dTTP with dUTP. These cDNA pieces possess a sole ‘A’ base and afterwards, sequencing adapters were ligated to the ends. After treatment with uracil DNA glycosylase (UDG), the merger of dUTP quenches the second strand during amplification. Products were enriched by PCR to establish the final cDNA library. Fragment size distribution was measured with an Agilent 2100 Bioanalyzer, and libraries were quantified and evaluated for quality with real-time quantitative PCR (RT-qPCR) using TaqMan probes. Valid libraries were sequenced end-to-end on the BGISEQ-500/MGISEQ-2000 system (BGI-Shenzhen, Shenzhen, China).

### 2.5. Statistical Analysis

Analysis of RNA-sequencing data was conducted using the ‘Dr. Tom’ system (https://biosys.bgi.com, accessed on day 1 January 2022). The expression level of the gene was calculated by RSEM (v1.3.1) [14]. The heatmap was drawn by pheatmap (v1.0.8) [Raivo Kolde. Package ‘pheatmap’. 4 January 2019 13:50:12 UTC.] according to the gene expression difference [15]. GO (http://www.geneontology.org/, accessed on 1 May 2022) and KEGG (https://www.kegg.jp/, accessed on 1 May 2022) enrichment analysis was performed by Phyper (https://en.wikipedia.org/wiki/Hypergeometric_distribution, accessed on 1 May 2022) based on a hypergeometric test. The significant levels of terms and pathways were corrected by Q value with a rigorous threshold (Q value ≤ 0.05). Results of the testosterone assay were compared with Student’s *t* test using SPSS ver20 statistics software (IBM, Armonk, NY, USA). A *p* < 0.05 was considered statistically significant. Moreover, the inter-assay is less than 15%; the intra-assay is less than 10%; and the minimum detection concentration is less than 0.1 ng/mL.

## 3. Results

### 3.1. Dietetic Obese Monkeys Showed Abnormal Testes Morphology

The results of morphological analysis showed that the seminiferous tubules in the control group were normal. The seminiferous cells were closely arranged, and the structure was clear. The spermatogonia, primary spermatocytes, spermatids, and other cells at various developmental stages could be observed in the seminiferous epithelium. In the obesity group, however, the structure of the spermatogenic epithelium was severely disordered and atrophic. The number of spermatogenic cells was reduced compared to the control, and the thickness of the spermatogenic epithelium was significantly thinner. The connection between Sertoli cells and spermatogenic cells was disrupted and loosely arranged. Only a small number of spermatozoa were present in the lumen of the seminiferous tubules (Figure 1).

### 3.2. The Testes Transcriptome of Obese Monkeys Differed Significantly from That of the Control Group

RNAs from the two groups of monkey testes were sequenced by RNA seq. With adjusted *p* values set to <0.05 and log2 [FC] > 1, we identified 2527 differentially abundant genes, including 1034 up-regulated genes and 1493 down-regulated genes (Figure 2A). When adjusted *p* values were set to <0.01 and log2 [FC] > 2, we extracted 507 differentially abundant genes, including 163 up-regulated genes and 344 down-regulated genes (Figure 2B). Using the transcripts per million (TPM) of 507 differentially abundant genes to draw a volcano map (Figure 2C) and heat map (Figure 2D), the results showed clear differences between obese and control animals. Key driver analysis (KDA) was utilized to identify the most important genes between the two groups, and ten driving genes, including interferon regulatory factor 1 *(IRF1)*, interferon regulatory factor 6 *(IRF6)*, homologous to the E6-AP carboxyl terminus (HECT) and regulator of chromosome condensation 1 (RCC1)-like domain (RLD) containing E3 ubiquitin protein ligase family member 5 (*HERC5*), HECT and RLD containing E3 ubiquitin protein ligase family member 6 (*HERC6*), interferon induced with helicase C domain 1 (*IFIH1)*, interferon induced protein with tetratricopeptide repeats 2 (*IFIT2)*, interferon induced protein with tetratricopeptide repeats *5 (IFIT5)*, interferon induced protein 35 (*IFI35)*, radical S-adenosyl methionine domain containing 2 (*RSAD2)*, Ubiquilin like (*UBQLNL)*, six initial genes and fifty-two associated genes were found (Figure 2E).

### 3.3. Gene Ontology and KEGG Pathway Analysis of the Differentially Abundant Genes between the Two Groups

GO and KEGG pathway analysis of the 163 up-regulated genes and 344 down-regulated genes was conducted. Among the 163 up-regulated genes, the top ten cellular components (CCs) included anchored components of membranes, extracellular regions, extracellular spaces, anchored components of the plasma membrane, eukaryotic 80 S initiation complex, integral component of the membrane, membranes, Coding Region Determinant (CRD)-mediated mRNA stability complex, cell surface, and pi-body (Figure 3A). Among the 344 down-regulated genes, the top ten CCs included stress fibers, contractile fibers, perinuclear region of cytoplasm, myosin complex, cell–cell contact zone, actin cytoskeleton, cytoplasm, heterotrimeric G-protein complex, Actin-related proteins-2/3 (Arp2/3) protein complex, and septin cytoskeleton (Figure 3B).

Among the 163 up-regulated genes, the top ten biological processes (BPs) were enriched in basement membrane organization, DNA methylation involved in gamete generation, cation transport, inhibition of neuroepithelial cell differentiation, methionine catabolic process, deoxyribonucleoside monophosphate catabolic process, polarity specification of proximal/distal axes, regulation of mRNA stability involved in response to stress, 4-nitrophenol metabolic process, pallium cell proliferation in forebrain, and P granule organization (Figure 4A). Among the 344 down-regulated genes, the top ten BPs were enriched in the defense response to the virus, interleukin-27-mediated signaling pathway, neutrophil aggregation, positive regulation of protein autophosphorylation, muscle filament sliding, positive regulation of smooth muscle cell differentiation, regulation of protein tyrosine kinase activity, regulation of hematopoietic stem cell proliferation, neutrophil chemotaxis, and leukocyte migration involved in the inflammatory response (Figure 4B).

Among the 163 up-regulated genes, the top ten molecular functions (MFs) were enriched in small molecule binding, copper ion binding, retinal isomerase activity, endochitinase activity, 4-nitrophenol 2-monooxygenase activity, voltage-gated proton channel activity, all-trans-retinyl-palmitate hydrolase, 11-cis retinol forming activity, all-trans-retinyl-ester hydrolase, 11-cis retinol forming activity, deoxyribonucleoside 5′-monophosphate N-glycosidase activity, and type I activin receptor binding (Figure 5A). Among the 344 down-regulated genes, the top ten MFs were α-1,3-mannosylglycoprotein 4-β-N-acetylglucosaminyltransferase activity, GTP binding, microfilament motor activity, GTPase activity, GDP binding, 2′-5′-oligoadenylate synthetase activity, C-X-C motif chemokine receptor 3 (*CXCR3)* chemokine receptor binding, actin binding, phospholipid scramblase activity, and actin filament binding (Figure 5B).

Among the 163 up-regulated genes, the top 20 KEGG pathways were enriched in glycosylphosphatidylinositol (GPI)-anchor biosynthesis, various types of N-glycan biosynthesis, Th1 and Th2 cell differentiation, viral protein interaction with cytokine and cytokine receptor, cysteine and methionine metabolism, graft-versus-host disease, steroid hormone biosynthesis, type I diabetes mellitus, chemical carcinogenesis-DNA adducts, signaling pathways regulating pluripotency of stem cells, arachidonic acid metabolism, and others (Figure 6A). Among the 344 down-regulated genes, the top 20 KEGG pathways included coronavirus disease-COVID-19, chemokine signaling pathway, thyroid hormone signaling pathway, long-term depression, ribosome, Interleukin (IL)-17 signaling pathway, circadian entrainment, advanced glycation end products- receptor for advanced glycation end products (AGE-RAGE) signaling pathway in diabetic complications, tyrosine metabolism, RIG-I-like receptor signaling pathway, estrogen signaling pathway, and others (Figure 6B).

### 3.4. Abnormal Expression of Infertility-Related Genes and Down-Regulation of Testosterone in Obese Monkeys

The 507 differentially abundant genes showed an intersection with genes related to the male sterility phenotype as deduced from the mouse genome informatics (MGI) database (Figure 7A). In the MGI database, 1014 genes were related to the male sterility phenotype, and among those genes, tudor domain-containing 5 (*TDRD5)*, chloride voltage-gated channel 2 (*CLCN2)*, mcrorchidia1 (*MORC1)*, regulatory factor X8 (*RFX8)*, spermatogenesis and oogenesis-specific basic helix-loop-helix 1 (*SOHLH1)*, interleukin 2 receptor subunit beta (*IL2RB)*, multi-ciliate differentiation and DNA synthesis-associated cell cycle protein *(MCIDAS)*, and zona pellucida binding protein *(ZPBP)*, were abnormally highly expressed in obese monkeys (Figure 7B), while eight genes, nuclear factor I A (*NFIA)*, protein-tyrosine phosphatase non-receptor type 11 (*PTPN11)*, TSC22 domain family member 3 (*TSC22D3)*, mitogen-activated protein kinase 6 (*MAPK6)*, phospholipase C beta 1 (*PLCB1)*, defective in cullin neddylation 1 domain containing 1 (*DCUN1D1)*, lipin 1 (*LPIN1)*, and glycine amidinotransferase (*GATM)*, were abnormally under-expressed (Figure 7C). Testosterone in the testes was measured in the two groups and was significantly lower in obese monkeys than controls (Figure 7D).

## 4. Discussion

Many important findings about reproduction, fertility, and development were based on experiments with mice, rats, and rabbits. However, human physiology and metabolism differ in many ways from that in these animals, which has limited the development of better treatments for sterility, infertility, and other problems caused by obesity [16]. Nonhuman primates are more similar to human beings in anatomy, physiology, and biochemistry than rodents are, and they have been used to study human reproductive physiology, especially in the composition, proliferation, and differentiation of gamete cells, and are considered as an ideal model for studying human reproductive physiology [17,18,19,20]. Important achievements have been made in male contraception based on monkey experiments [21,22]. Azoospermia, which has been described in nonhuman primates, is also more similar to the human condition [23,24]. Monkeys have an important value and incomparable advantages over other species, and there have not been enough studies on obesity-related infertility or reproductive decline in monkeys, which has limited the development of drugs for controlling those diseases. In this study, we found that the pathological morphology of the testes of cynomolgus monkeys was very similar to the testicular pathology of obese and infertile humans. The structure of the seminiferous epithelium in testes from obese individuals showed significant disorder, and the walls of the seminiferous tubules were noticeably thinner. The number of spermatogenic cells was reduced; the protective connection between the supporting Sertoli cells and the spermatogenic cells was broken; the arrangement was loose; and the integrity of the BTB was also damaged. Spermatogenesis and the BTB integrity depend on a high level of testosterone in the testes [25,26]. Previous studies reported that obesity led to hormone dysfunction. The hypothalamic pituitary gonadal axis (HPG) negative feedback pathway was disrupted, which resulted in decreased testosterone levels [27]. Our results showed that the testosterone in the testes of obese monkeys was significantly lower than that of the normal weight control group. This is similar to the disorders of the testosterone regulatory axis seen in obese humans.

The results of transcriptome analysis showed a significant change in gene expression in the testes of obese monkeys. Key driver analysis showed that ten genes were closely related to immunity or inflammatory processes, which suggested that dietetic obesity could lead to chronic inflammation of the testes. This condition could be one pathogenic mechanism leading to infertility in obese people. Using relatively strict threshold screening, 163 up-regulated genes and 344 down-regulated genes were identified in testes from obese monkeys. The biological process with the highest score among the 163 up-regulated genes is basement membrane organization, which is closely related to the BTB, suggesting that obesity may cause damage to the BTB and the dysfunction of spermatogenesis by negatively affecting the expression of regulatory genes related to basement membrane organization. Enterogenic endotoxin production by intestinal microbiota, which can be induced by a high-fat or high-sugar diet, can activate toll-like receptors (TLRs) related to inflammation on testicular microvascular endothelial cells and release inflammatory factors [28]. Inflammatory cytokines infiltrate the testicular parenchyma through the damaged microvascular endothelial cells, and inflammatory cells gather in the testis stroma, causing inflammation, damaging the spermatogenic cells, and resulting in male reproductive dysfunction [29]. In this study, the biological processes of differentially abundant genes were mostly enriched in immune and inflammatory responses, and also in related KEGG pathways and key driver genes, suggesting that the infertility associated with male dietetic obesity may have close links to the inflammatory process.

The DNA methylation level of spermatozoa showed a dynamic change different from that of somatic cells during spermatogenesis [30]. The methylation of male DNA starts before maturation of the fetal gonads and ends in the process of sperm maturation [31]. Demethylation takes place twice in one’s life. The first time is when re-establishing the imprint in the period of forming mature gametes, and the second wave is when the imprint is replaced by a new imprint pattern, which occurs during embryonic development [31]. Abnormal DNA methylation is closely related to male infertility [32]. Abnormal methylation of sperm DNA may be an important reason for early spontaneous abortion or fetal termination [33]. Metabolic abnormalities and decreased fertility in offspring of dietetic obese male mice are considered to be related to changes in sperm DNA methylation [34,35]. Paternal obesity could change DNA methylation patterns in sperm, and the offspring of an obese father had an abnormal methylation pattern in the differentially methylated region (DMR) of imprinted genes [11,35,36]. This study revealed that the DNA methylation process involved in gamete generation was significantly enriched, suggesting that dietetic obesity might be related to abnormal spermatogenesis through alterations in DNA methylation.

*TDRD5* protein is essential for pachytene piRNA biogenesis and may play an important role in male infertility. Previous studies found that *TDRD5* was abnormally expressed in the testes of individuals with azoospermia [37,38]. *SOHLH1* is a testis-specific basic helix-loop-helix transcription factor which is essential for spermatogonial differentiation. Mutations in the *SOHLH1* protein resulted in non-obstructive azoospermia [39,40]. *MCIDAS* is necessary for the generation of functional multi-ciliated cells in the efferent ducts that are required for spermatozoa to enter the epididymis [41]. Mutations in *MCIDAS* reduced the generation of multiple motile cilia and caused male infertility [41]. *ZPBP* is localized to the acrosome in human spermatozoa, and males lacking *ZPBP1* were sterile, with abnormal round-headed sperm lacking forward motility [42]. *TSC22D3* is a widely expressed dexamethasone-induced transcript that has been proposed to be important in immunity and adipogenesis. *TSC22D3*-deficient males were infertile and exhibited severe testicular dysplasia and a high number of apoptotic cells within the seminiferous tubules [43]. *MORC1* is expressed in germ cells (spermatogonia and spermatocytes) of the testis [44]. Besides spermatogenesis, *MORC1* is also essential for the facilitation of DNA methylation and transposon repression in the male embryonic germ cells. The finding showed that mutations in *MORC1* caused the loss of male-specific germ cells and infertility in mice [45]. The NFI (nuclear factor I) family is mainly involved in regulating the development of stem cells. *NFIX*, one of the members, is expressed in spermatocytes and plays an important role in spermatogenesis. The deficiency of *NFIX* resulted in multinucleation in spermatocytes, structural defects in the synaptonemal complex, etc., which ultimately led to the inhibition of spermatogenesis [46]. *PTPN11*, also known as *SHP2* (SH2-containing tyrosine phosphatase 2), is essential for the self-renewal of spermatogonial stem cells and the production of other male germ cells [47]. It is also associated with the integrity of the BTB [48]. Mutations in *PTPN11* resulted in Noonan and LEOPARD syndromes, both of which manifested reproductive defects in the reproductive system such as male infertility [49,50]. *PLCB1* is a vital regulator of the acrosome reaction in spermatozoa. Mutations in *PLCB1* reduced the rate of acrosome reaction, fertilization rate, and the probability of embryo development to mulberry or blastocyst [51], resulting in a reduction in male fertility. *DCUN1D1*, also known as *SCCRO* (squamous cell carcinoma-related oncogene), is an important regulator of neddylation in mammals. An animal study found various abnormal morphologies that appeared in the spermatozoa of *DCUN1D1*-/- mice, such as macrocephaly and multiple flagella. In addition, these aberrant spermatozoa indicated the presence of abnormal intercellular bridges, which were probably responsible for specific infertility and inability to release mature spermatozoa of *DCUN1D1*-/- mice [52]. Here, we found that the genes *TDRD5*, *SOHLH1*, *MCIDAS*, and *ZPBP* were abnormally expressed in the testes of obese monkeys compared with normal-weight controls, suggesting that the dysfunction of those genes and their related biological functions should be given prominence in studies of human obesity-induced male infertility.

Despite the rapid development of multiomics, research on obesity or the effects of a high-calorie diet on the testicular transcriptome have mainly been confined to rats and mice with few studies focused on humans or non-human primates. In our research, we investigated the influence of a typical modern diet, high in fat and fructose, and cholesterol-rich, on gene expression in the testes of cynomolgus monkeys, which afforded valuable insights into the effects of diet and obesity on male infertility.

## 5. Conclusions

Dietetic obese monkeys showed abnormal testicular morphology, and their testes transcriptome was significantly altered. The differentially abundant genes were predominantly associated with blood–testes barrier function, immunity, inflammation, DNA methylation involved in gametogenesis, decreased testosterone level, and abnormal expression of infertility-related genes. This work will provide an important basis for future research and the development of treatments for male infertility caused by obesity.

## Figures and Tables

**Figure 1 genes-14-00557-f001:**
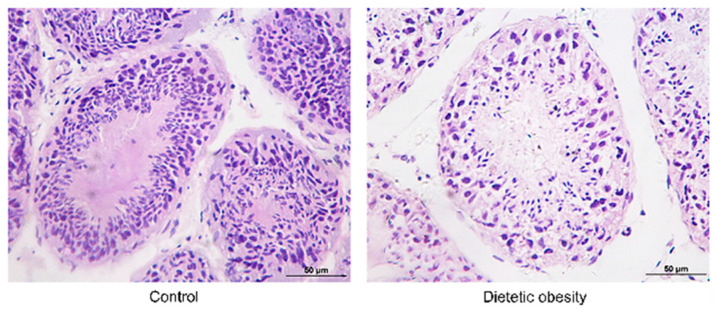
Morphological analysis of testis from control and dietetic obese monkeys. Scale bar = 50 um.

**Figure 2 genes-14-00557-f002:**
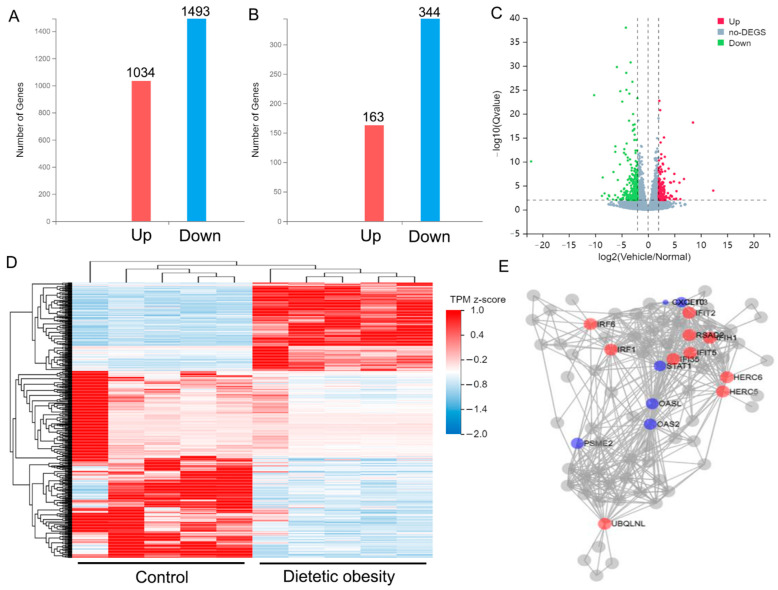
Differentially abundant genes between the dietetic obese monkeys and control group. (**A**) Differentially abundant genes with adjusted *p* value < 0.05 and log2 [FC] > 1; (**B**) differentially abundant genes with adjusted *p* value < 0.01 and log2 [FC] > 2; volcano map (**C**) and heat map (**D**) of the 507 differentially abundant genes in (**B**,**E**) visual network of key driver analysis.

**Figure 3 genes-14-00557-f003:**
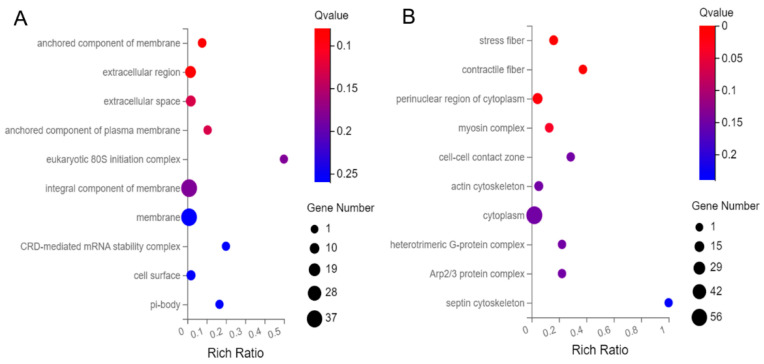
Bubble diagram of top 10 Cellular Components (CC) for the 507 differentially abundant genes. (**A**) CC of the up-regulated genes, and (**B**) CC of the down-regulated genes.

**Figure 4 genes-14-00557-f004:**
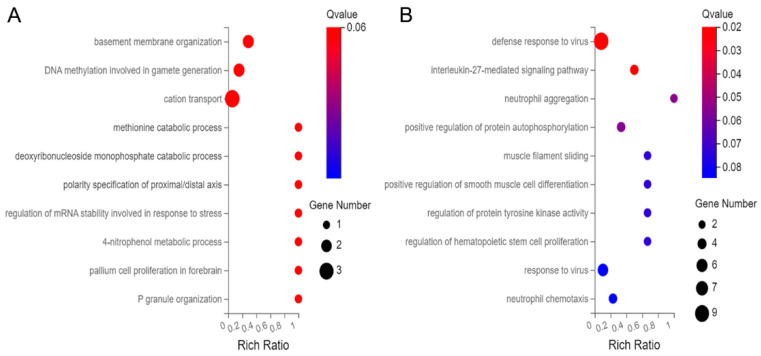
Bubble diagram of top 10 Biological Processes (BP) for the 507 differentially abundant genes. (**A**) BP of the up-regulated genes, and (**B**) BP of the down-regulated genes.

**Figure 5 genes-14-00557-f005:**
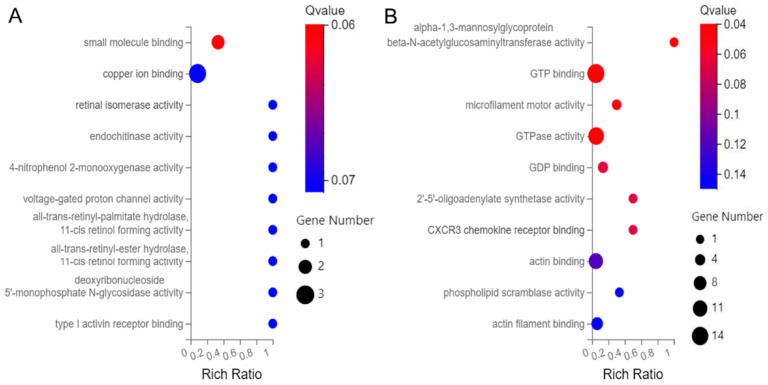
Bubble diagram of top 10 Molecular Function (MF) for the 507 differentially abundant genes. (**A**) MF of the up-regulated genes, and (**B**) MF of the down-regulated genes.

**Figure 6 genes-14-00557-f006:**
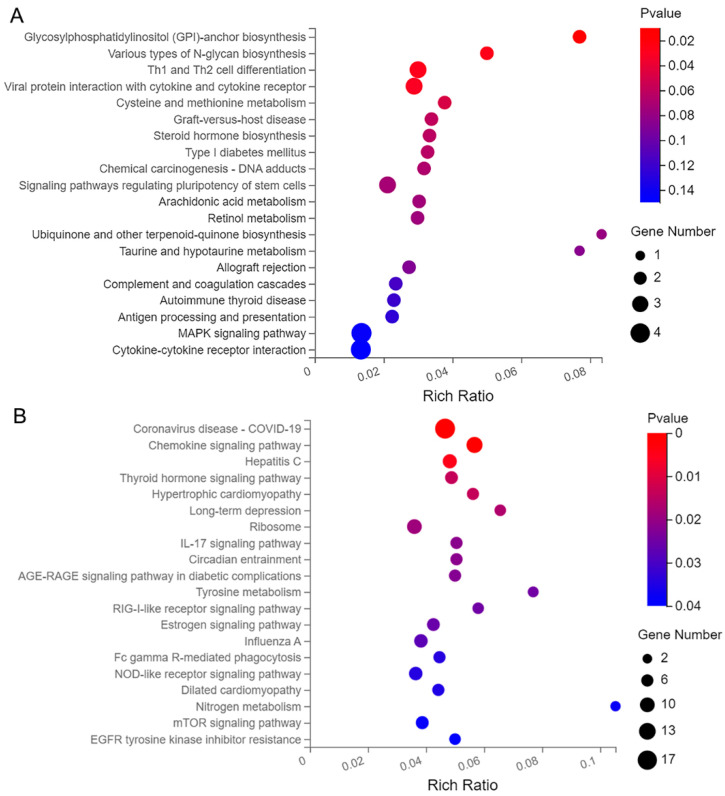
Bubble diagram of top 20 KEGG pathway for the 507 differentially abundant genes. (**A**) KEGG pathway of the up-regulated genes, and (**B**) KEGG pathway of the down-regulated gene.

**Figure 7 genes-14-00557-f007:**
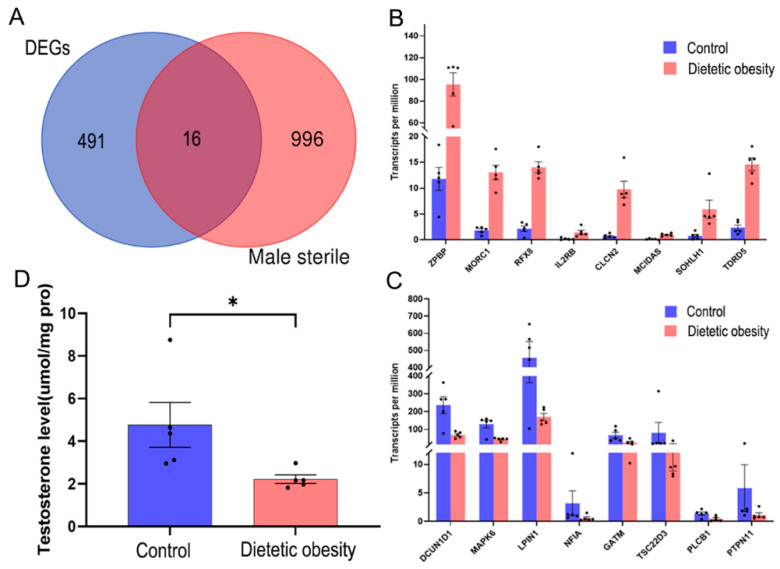
Abnormal expression of infertility related genes and low level of testosterone in dietetic obese monkeys. (**A**) Venn diagram of the 507 differentially abundant genes and genes related to male sterility phenotype in mouse genome informatics database; genes related with sterility higher (**B**) or lower (**C**) in dietetic obese monkeys than the control. (**D**) Testosterone in testis between the two groups, and the * above the bars indicates significant difference (*p* < 0.05).

## Data Availability

Data is available upon request to the corresponding author.

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
