# Peer review of "Effect of Dietetic Obesity on Testicular Transcriptome in Cynomolgus Monkeys"

_genes, 2023, doi:10.3390/genes14030557_

Round 1
Reviewer 1 Report
This manuscript from Zhang et al. reports the effect of dietetic obesity on testes morphology and transcriptome. These results could be very interesting for the treatment of male infertility caused by obesity. However, some concerns are as follows.
1.- Several essential information regarding the animal housing, handling and welfare is missing, especially considering the use of primates. The description of animal housing and welfare should allow the reproducibility of the results. Moreover, the lack of essential information regarding animals and the experimental design does not allow the critical appraisal of evidence from this study. Thus, I very strongly recommend using the SYRCLE’s RoB tool (DOI: 10.1186/1471-2288-14-43) to provide further essential information regarding animal housing, handling, and experimental design. As an example, some essential information that is not present in this manuscript is (i) the blindness during the housing, intervention, and outcome assessment, (ii) the information of animals in each group (age, weight, etc), (iii) the specific housing conditions of monkeys (temperature, humidity, hours of light, environmental enrichment, etc), (iv) the experimental design, among others.
2.- Regarding the previous comment, it is essential to provide the reference of the approved report of the ethic committee within Material and Methods (L69). Moreover, the authors should provide a translated copy (in English) of the ethic committee report approving the study to the editor and the reviewers. Please provide this document.
3.- The specific composition of food in the control and high fat diet groups should be specified and provided in a supplementary file. The intervention time in which both groups (control and high fat diet) must also be provided. Please provide this information.
4.- L48. The accurate term for sperm should be “reduced motility” instead of “poor movement”. Please revise.
5.- L63. The accurate term for “American diet” should be “Western pattern diet”. Please revise.
6.- The authors should specify the testicular portion intended for RNA-sequencing. Since the testis are not completely homogeneous (the dorsal part contains different cell types than the ventral part), it is essential to specify the exact location of the testicular tissue recovery.
7.- L129-L130. Spermatogonia, primary spermatocytes, spermatids and other cells at various developmental stages cannot be observed in the lumen but in the seminiferous epithelium. Please revise.
8.- Data analysis of RNA-sequencing should contain further details. Please provide all the essential information and key points of this analysis.
9.- In the author contributions, no author has been specified to do animal handling nor laboratory analysis. Please provide this information.
Reviewer 2 Report
The manuscript entitled "Effect of dietetic obesity on testicular transcriptome in cynomolus monkeys" is very interesting transcriptomic investigation of differentially expressed genes in testis related to obesity. The obtained results could be used for better understanding of obesity and its mechanism involved in prevention and cure male sterility. The manuscript is well organized, the cited literature is relevant to the questions discussed and the figures are well made. However, I am sending a few comments below, which in my opinion should be taken into account by the authors:
1. There is not enough information about animals and experimental conditions. For instance, I think that it is important to know the age of the monkeys, the time period of the diet-induced obesity, assessment of obesity and testicular sampling methods!
2. Line 55 – what did you mean by ”… proteome and ect al of gametes…”; is it a spelling error?
3. Line 129-130 you say: ”The spermatogonia, primary spermatocytes, spermatids and other cells at various developmental stages could be observed in the lumen”. As I can see in figure 1. in control group all those cells are in adluminal compartment and only a few spermatozoa are in lumen. Furthermore, at line 148-149 you say that this pathological morphology was observed in the obesity group: ”Large numbers of spermatogenic cells had been translocated to the lumen…”. Please provide the correct version!
Reviewer 3 Report
Review
Introduction:
The authors investigated the effects of a high-fat-carbohydrate-high-carb diet on the transcriptome in monkey nuclei.
Current medical findings provide a lot of disturbing information about the fertility of men with a BMI > 30. Above all, this group of patients suffers from numerous fertility disorders. The cause of the malfunction of the male reproductive system is the negative impact of adipose tissue on the endocrine system, metabolism and health. Changes in sperm parameters are observed in overweight men: abnormal sperm concentrations and reduced sperm motility. Studies by Kort et al. (2006) showed a negative correlation between BMI and sperm motility. In men of normal body weight, 18. 6 million mobile sperm were reported, in obese men only 3. 6 million and in men with a BMI > 30 only 0. 7 million. Sperm chromatin integrity was also investigated by flow cytometry and increased DNA fragmentation of sperm was observed by calculating the DNA Fragmentation Index (DFI). Overweight men are often diagnosed with a specific hormonal profile characterized by hypogonadism, hyperestrogenism and hypoandrogenism. Therefore, a decrease in androgen levels, which is often proportional to the degree of obesity, a decrease in gonadotropins, a decrease in inhibitor B and an increase in oestrogen levels are observed. Increased E2 concentrations influence the number of GnRH pulses and the pituitary gland that regulates the body’s own gonadotropins (FSH and LH). Increased E2 concentrations in overweight men are thought to reduce FSH and LH production, resulting in decreased testicular function and decreased testosterone production. An excess of estrogen in the male body also negatively affects the process of spermatogenesis. Decreased levels of the sexual globulin SHGB, which binds the sex hormones, are also observed in this group of patients. It is believed that a disturbed endocrine system can alter sperm parameters in an overweight man. Excessive consumption of highly processed foods, meat and meat products may lead to a significant increase in total fat consumption and saturated fatty acids, which may affect fertility. In the study Jensen and co-author. (2013) investigated the relationship between fat intake and the quality of sperm parameters. Men who consumed saturated fatty acids at least 15. 2% of the energy value of their usual diet were characterised by poorer sperm parameters, including a 38% lower sperm concentration and a 41% lower total sperm count compared to men who consumed saturated fatty acids at less than 11. 2% of the energy value of their diet. ingested food.
In men, the frequent consumption of processed foods and thus of easily digestible simple carbohydrates leads to an increase in blood sugar levels and the development of insulin resistance, which may lead to an increase in oxidative stress, which also has a negative impact on sperm quality. The function of the hypothalamic pituitary-gonad axis, which is responsible for spermatogenesis, may be impaired by high glucose and insulin concentrations.
Notes on the details:
Article title and introduction are not objectionable
Material and methodology
The experimental set-up and the methods of analysis used are:
Explanation and supplementation require:
What was the average weight of the control and experimental groups at the beginning of the experiment and at the end of the experiment?
How long did the experience last?
This information should be included in the working methodology.
Results
The chapter is well written, with seven diagrams.
The authors found that the testicular transcriptome of obese monkeys differed significantly from that of the control group, which I think is one of the most important findings of this work. The second important fact was that a significantly lower testosterone level was found in obese males. It is unfortunate that, in addition to testosterone levels, the authors did not determine the luteinizing hormone (LH), which is responsible for the production of testosterone, and the follicle stimulating hormone (FSH), which is responsible for the course of spermatogenesis, in order to give a more complete picture of the nutritional disorders.
The chapter “Discussion” is unobjectionable, the authors refer to all the results of their studies.
Summary:
Dietary overweight monkeys showed abnormal testicular morphology and their testicular transcriptions were greatly altered. DEGs have been mainly associated with blood test barrier function, immunity, inflammation, DNA methylation involved in gametogenesis, decreased testosterone levels and abnormal expression of genes associated with infertility. This work will provide an important basis for future research and development of therapies for male infertility due to obesity.
In general, this study may summarize that the pathological testicular morphology of cynomolgus monkeys is very similar to testicular pathology of obese and infertile humans.
After considering the comments, the work can be published in Genes.
Round 2
Reviewer 1 Report
The Authors addressed my concerns and comments.
Author Response
Thank you.